# Programmed Cell Death Protein-1 Regulation in Response to SARS-CoV-2 in Paediatric Multisystem Inflammatory Syndrome Temporally Associated with SARS-CoV-2: A Prospective Cohort Study

**DOI:** 10.3390/ijms25115968

**Published:** 2024-05-29

**Authors:** Violetta Opoka-Winiarska, Ewelina Grywalska, Izabela Morawska-Michalska, Izabela Korona-Głowniak, Olga Kądziołka, Krzysztof Gosik, Adam Majchrzak, Mansur Rahnama-Hezavah, Paulina Niedźwiedzka-Rystwej

**Affiliations:** 1Department of Pediatric Pulmonology and Rheumatology, Medical University of Lublin, 20-093 Lublin, Poland; violetta.opoka-winiarska@umlub.pl; 2Department of Experimental Immunology, Medical University of Lublin, 20-093 Lublin, Poland; ewelina.grywalska@gmail.com (E.G.); krzysztof.gosik@umlub.pl (K.G.); 3Department of Clinical Immunology, Medical University of Lublin, 20-093 Lublin, Poland; izabelamorawska19@gmail.com; 4Department of Pharmaceutical Microbiology, Faculty of Pharmacy, Medical University of Lublin, 20-093 Lublin, Poland; iza.glowniak@umlub.pl; 5Department of Paediatric Pulmonology and Rheumatology, University Children’s Hospital of Lublin, 20-093 Lublin, Poland; olga.kadziolka@uszd.lublin.pl; 6Department of Infectious, Tropical Diseases and Immune Deficiency, Pomeranian Medical University in Szczecin, 71-455 Szczecin, Poland; adam.majchrzak@pum.edu.pl; 7Chair and Department of Oral Surgery, Medical University of Lublin, 20-093 Lublin, Poland; mansur.rahnama-hezavah@umlub.pl; 8Institute of Biology, University of Szczecin, 71-412 Szczecin, Poland; 9Center for Experimental Immunology and Immunobiology in Infectious Diseases and Cancer, University of Szczecin, 71-412 Szczecin, Poland

**Keywords:** programmed cell death protein 1, Paediatric Multisystem Inflammatory Syndrome, severe acute respiratory syndrome coronavirus 2, immunophenotype, lymphocyte

## Abstract

The role of programmed death cell protein 1 (PD-1) has already been described in a range of various diseases, including COVID-19. This study provides new, innovative data, related to the expression of PD-1 and the risk of Paediatric Inflammatory Multisystem Syndrome, temporally associated with SARS-CoV-2 infection (PIMS-TS)—a rare, but potentially life-threatening complication of COVID-19. In this study, we evaluated the expression of PD-1 protein in patients with PIMS. Blood samples were taken from patients at the time of diagnosis (n = 33), after 6 weeks (n = 33), 3 months (n = 24), 6 months (n = 24) and 12 months (n = 8). The immunophenotypes were evaluated in flow cytometry. The control group consisted of 35 healthy children with negative SARS-CoV-2 antigen/PCR test, who were asymptomatic and had no history of allergic, autoimmune or oncological diseases. The associations between immunophenotypes, biochemical findings and clinical data were analysed. Significant increases in the expression of PD-1 for CD4+ and CD8+ T cells, compared to the control group, were observed in the day of admission, with a gradual decrease during the first weeks from initiation of treatment. This study sheds new light on the pathogenesis of PIMS-TS, emphasizing the role of PD-1 protein. Future research is essential for early risk prediction in SARS-CoV-2 patients and for devising effective clinical prevention and management strategies.

## 1. Introduction

Paediatric Multisystem Inflammatory Syndrome temporally associated with SARS-CoV-2 (PIMS-TS) is a life-threatening complication that typically occurs around two to six weeks after SARS-CoV-2 infection in children and young people [1]. The disease was described during the first wave of the COVID-19 pandemic, in 2020 [2,3]. The clinical picture of the disease has been well characterized. The main symptom of the disease is fever, but the severe course of the disease is determined by organ involvement: heart, lungs, gastrointestinal tract, often coagulation disorders and others [4,5,6]. Immunosuppressive therapy with intravenous immunoglobulins, glucocorticoids, antiplatelet or anticoagulant drugs is effective in most cases [7]. However, the pathogenesis is partially understood and the long-term effects of PIMS-TS are still being carefully monitored. The impact of SARS-CoV-2 infection on the immune system of children with PIMS-TS remains unclear and requires explanation. The importance of immune checkpoints in balancing the immune response in COVID-19 is also considered in research [8].

Programmed cell death-1 protein (PD-1 or CD279) is a member of the CD28 superfamily, expressed on CD4+ and CD8+ T cells, B cells, NK cells, and activated monocytes. Binding to the ligands PD-L1 and PD-L2 on immunological cells inhibits their function to regulate T cell activation and prevents immune-mediated tissue damage. Our recently published study showed that the humoral immune response to SARS-CoV-2 infection in healthy children and in patients with juvenile idiopathic arthritis (JIA) relates to persistent activation of PD-1 expression [9]. 

The aim of the study was to evaluate the expression of selected suppressive molecules in PIMS-TS patients in relation to laboratory parameters at the time of diagnosis and annual follow-up. The presented results of our observation are intended to show new data in the pathogenesis of the disease in prospective observation. Understanding the pathogenesis is important for determining effective management and possibly prevention in the future. 

## 2. Results

The results of the study have been collected in Table 1 and Table 2. 

### Clinical Characteristic of Patients with PIMS

The study was carried out during the first wave of the COVID infection when there was no widespread availability of the COVID vaccine among the paediatric population. The analysed group included 33 patients diagnosed with PIMS compared to 35 healthy children in the control group. A detailed individual medical interview was conducted. No previous diseases associated with immune response disorders were detected. PIMS-TS was diagnosed based on current criteria [7,10]. The groups were balanced for age. Within the observation group, all patients were treated with intravenous immunoglobulins (IVIGs) and acetylsalicylic acid (ASA), while 39.4% were treated with glucocorticoids (GCs) in accordance with Polish recommendations [11] (Table 1). The number of boys and girls in the entire study group was 19/14, while in the control group, it was 16/19. 

At the time of disease diagnosis, patients with PIMS-TS had a significantly higher value of erythrocyte sedimentation rate (ESR, median: 65.5 mm/h, IQR: 54.0–120.0 mm/h vs. median: 6.0 mm/h, IQR: 2.0–16.0 mm/h, *p* < 0.0001) and C-reactive protein (CRP, median: 8.9 mg/dL, IQR: 4.7–22.2 mg/dL vs. median: 0.06 mg/dL, IQR: 0.06–0.08 mg/dL, *p* < 0.0001). Next, the concentration of IgG anti-SARS-CoV-2 antibodies was higher in patients with PIMS, compared to controls, with *p* value considered on the borderline of statistical significance (median: 316 ng/mL, IQR: 155–508 ng/mL vs. median: 97.1 ng/mL, IQR: 53.2–417.0 ng/mL, *p* = 0.09) (Table 2).

Subsequent visits with the control of laboratory tests included patients who came for a visit on the date specified in the plan. Therefore, the last follow-up in this study included only eight patients. The patients were in good condition, no disease symptoms were observed in the medical history and physical examination. In general, further follow-up was recommended for all children after PIMS-TS.

Patients were treated in accordance with Polish recommendations [11]. First-line treatment was intravenous immunoglobulin (IVIG) infusion at a dose of 2.0 g/kg. Indications for the use of glucocorticoids included severe or deteriorating general condition of the child; characteristics of shock; lack of improvement, in particular, persistent fever 24–36 h after the end of the IVIG infusion. No child experienced coronary artery aneurysms or anaphylactic reactions after receiving IVIG. Treatment with GC was used for 3 to 6 weeks.

## 3. Discussion

Paediatric Inflammatory Multisystem Syndrome (PIMS), temporally associated with SARS-CoV-2 infection (PIMS-TS), known also as Multisystem Inflammatory Syndrome in children (MIS-C), is a rare, but life-threatening complication of COVID-19. 

According to current knowledge, the symptoms of PIMS-TS occur as a result of both humoral and cellular disturbed immune responses. Cytokine profiling in PIMS-TS patients indicates myeloid cell chemotaxis and inflammation. An assessment of cytokines showed an increased release of the pro-inflammatory cytokines Il-1, IL-6, IL-18, IL-17A, TNF-α and chemokines, which regulate the recruitment and modulation of natural killer (NK) cells, neutrophils, monocytes and T lymphocytes. Pro-inflammatory cytokines play a key role in the pathogenesis of disease [12].

Nevertheless, the pathogenesis of PIMS-TS requires answering many questions. Our study suggests that the expression of programmed cell death protein-1 (PD-1) may play a pivotal role in the interruption of the immune system, as observed in PIMS. The increase in soluble programmed death-ligand 1 indicates exhaustion and immunosuppression, possibly reflecting a compensatory inflammatory response. 

PD-1 is expressed on various cells building the immune system, including not only lymphocytes, but also macrophages, dendritic cells, monocytes and myeloid cells. Binding to specific ligands (PD-L1, PD-L2) leads to cell activation, cytokine release, shift in cell metabolism and eventually apoptosis, resulting in lymphocyte exhaustion. The role of PD-1/PD-L1 axis has been proved for the pathogenesis of multiple conditions, like inflammatory bowel disease, systemic lupus erythematosus, systemic vasculitis, type 1 diabetes and others. Growing data highlight its importance also in numerous infectious diseases, including human immunodeficiency virus (HIV), hepatitis B virus (HBV), hepatitis C virus (HCV) and SARS-CoV-2 [13]. Regarding COVID-19, the correlation between PD-1+ and PD-L1+ expression and disease severity has been recently investigated, giving an attractive biomarker of COVID-19 severity [14]. 

PIMS-TS is a serious complication of SARS-CoV-2 infection, which develops approximately 3–6 weeks after initial infection. While the exact mechanism remains unclear, immunological dysregulation leads to the uncontrolled release of multiple proteins, mainly interleukin-6 (cytokine storm), resulting in hyperinflammation, multiorgan injury and shock. Current management of PIMS-TS/MIS-C consists of supportive care and agents with a suppressive and/or modulatory impact on the immune system (systemic therapy with immunoglobulin and/or steroids) [11].

Increasing evidence confirms that the immune system is continuously regulated by multiple, multi-level regulatory mechanisms, including immune checkpoint proteins, especially PD-1. Precisely, the inhibitory function of PD-1 is also under complex regulation and depends on the interaction with SHP-2 phosphatase, resulting in multiple down-regulation pathways, including PI3K-PDK1-AKT-mTOR, RAS-RAF-MEK-ERK and JAKs-STAT, that are involved in numerous diseases [13].

Currently, studies regarding the impact of PD-1 expression in PIMS-TS are hardly available. According to data presented by Suksatan et al., children suffering from PIMS-TS have lower total T cell frequencies than healthy children, a similar CD4+ expression, and a lower CD8+ count than children with mild SARS-CoV-2 infection [15].

Our study suggests that a pivotal role of the clinical outcome of SARS-CoV-2 infection and the risk of complications, including PIMS-TS, is the dysregulation of immune balance, controlled by complexed network of factors, where the main role is played by PD-1 protein. The findings presented here show changes in certain biochemical parameters after COVID-19; their concentrations were elevated at the beginning of observation and decreased over time, which is consistent with recent knowledge. Results regarding the expression of PD-1 on CD4+ and CD8+ T cells show significant decreases during PIMS-TS/MIS-C development. 

Carcillo et al., in their study published 2.5 years before the first SARS-CoV-2 infections, provide information about “immune paralysis-associated multiple organ dysfunction syndrome” and highlight the possible complications of lymphocyte apoptosis. Moreover, authors suggest that full recovery from organ disfunction comes 6–18 weeks after epithelial, endothelial, mitochondrial, and immune cell regeneration [16].

Few studies describe the differences between patients with acute SARS-CoV-2 infections and patients with PIMS-TS, where CRP elevation and reduced platelet count were more pronounced in PIMS-TS than in acute COVID-19 patients [17]. It should be emphasized that PIMS is a late inflammatory reaction of the immune system 3–6 weeks after previous infection, when the virus is not detected. A similar sequence of events is observed in other inflammatory diseases such as rheumatic fever or reactive arthritis.

It should also be emphasized that our study was conducted in the first phase of the COVID-19 pandemic, when the variant Delta of COVID-19 virus was dominant. In the later phase of the pandemic, we observed a reduction in the incidence of PIMS-TS, although the disease still occurs. Observations by other authors also indicate that the number of PIMS-TS cases decreased with the Delta and/or Omicron variant of SARS-CoV-2. This might be explained by two theories: first, in the result of increasing seroprevalence, through infection and/or vaccination; secondly, because of mutations in the virus’ spike protein, affecting infectivity, virulence and disease outcomes. Together with other reports on PIMS-TS, it is believed that the disproportion of minority ethnic groups affected is possibly due to genetic factors, but the exact reason remains unclear. Klocperk et al. described few differences between patients with PIMS-TS and healthy post-COVID-19 children, where children with this complication were characterized with elevated IFN-γ activity, increased serum cytokine B cell activating factor (BAFF) and decreased proliferation-inducing ligand (APRIL) concentration [18]. In a similar study, Butters et al. found that children with PIMS-TS had a lower proportion of polyfunctional SARS-CoV-2-specific CD4+ T cells compared to healthy SARS-CoV-2 seropositive children, while the level of monofunctional SARS-CoV-2-specific CD4+ T cells was high in both groups [19]. 

Studies on changes in immunophenotype in patients with PIMS-TS are limited. Jaxybayeva et al. identified these changes in less than 6 months, 12 months, 24 months and more than 24 months after PIMS-TS, where apart from clinical, mainly cardiovascular differences, the immune profiling showed reconstitution of CD3+ CD4+ T-cells and NK-cells, reduction in CD19+ B cells, maintenance of high CD3+ CD8+ T-cells and expression of few other proteins, namely, CD3-HLA-DR+, CD25, CD279, and finally, CD95 [20]. 

Children with PIMS-TS had a decreased percentage of CD3+ cells, NK cells and Vδ1 compared to controls [21]. Rybkina et al. found similar results, with a suggestion of a potential role for tissue-derived T cells in the treatment of PIMS-TS. [22] The disease may also occur as a complication of SARS-CoV-2 infection in neonates (MIS-N) [23].

Deep immune profiling, performed by Redmond et al., tried to explain potential immunological mechanisms underlying the clinical heterogeneity of PIMS-TS. The percentage of SARS-CoV-2-specific CD4+ T-cells were elevated mainly in patients with neurological and respiratory involvement, suggesting predilection for these functions. Interestingly, decreased concentrations with some shock-associated serum biomarkers in patients with diarrhoea suggest that this complication may have protective effects [24].

Interestingly, the presented correlation between PD-1 expression and disease severity, as well as the development of complications like PIMS, may serve both as biomarkers for disease progression, but also as possible therapeutic targets by anti-PD-1 drugs, which have been used in various conditions, including cancers and few infectious diseases [14].

The administration of PD-1 inhibitors could ameliorate the lymphocyte exhaustion observed in PIMS-TS/MIS-C, and therefore gives a chance to avoid these complications. 

Nonetheless, the enthusiastic impact of potentially targeting SARS-CoV-2 or PIMS-TS has also a dark side, because blocking this pathway may give important complications. Therefore, future studies are essential for exploring the underlying mechanisms of PD-1 modulation in PIMS-TS and evaluating the efficacy and safety of target treatment. 

## 4. Materials and Methods

### 4.1. Study Group

All subjects were diagnosed and treated in the Department of Paediatric Pulmonology and Rheumatology, Medical University of Lublin (Poland). The criterion for inclusion in the study was the diagnosis of PIMS-TS. PIMS-TS was diagnosed based on current criteria [7,14].

The study was conducted in 33 patients at the time of diagnosis of PIMS-TS (before initiation of treatment), and subsequently after 6 weeks (33 patients), after 3 months (24 patients), after 6 months (24 patients), after 9 months (12 patients) and after a year (8 patients). The study was also carried out in 35 healthy children of the control group.

In total, 137 serum samples were obtained from patients who were admitted to our department from November 2020 to February 2022, while the incidence of PIMS-TS included in the study was from November 2020 to April 2022. The study (incidence) was conducted in the period prior to the introduction of COVID-19 vaccinations for the children aged before 18; as such, none of the subjects were vaccinated before PIMS-TS diagnosis and only 3 children received the COVID-19 vaccination by the end of the study.

The control group included 35 healthy children of health workers. We excluded children who were taking medication affecting the immune system; had reported symptoms of infection in the last three months before the study; or patients with diagnosed chronic diseases, such as allergies or inflammatory, autoimmune, or oncological diseases. Sera samples after 6 weeks and subsequent tests were obtained from patients during routine laboratory tests during one-day visits to the department. Clinical data were extracted from the electronic medical record.

Detection of specific antibodies was performed by ELISA-based tests for anti-SARS-CoV-2, IgA and IgG, designed and distributed by Euroimmun (Lubeck, Germany). Tests, directed against the S1 subunit/domain of the spike protein of SARS-CoV-2, were used according to the manufacturer’s instructions. Results were calculated as the absorbance value of the sample divided by the absorbance value of the calibrators and expressed as an extinction ratio. We utilized the manufacturer’s interpretation of the ratio with samples < 0.8 classified as no antibody present, 0.8–1.1 as indeterminate, and ≥1.1 as containing antibodies. 

Laboratory and immunological parameters were assessed using standard methods by a hospital laboratory. Flow cytometry was used to assess PD-1 receptor expression.

### 4.2. Flow Cytometry

After obtaining the appropriate consents, blood samples were taken from the patients into EDTA-coated tubes and immediately transported to the Department of Clinical Immunology. According to the protocol, 50ul of whole blood was stained with 5ul of each of the anti-human fluorescent conjugated antibodies (BD Biosciences, Franklin Lakes, NJ, USA): anti-CD45 FITC, anti-14 PE (simultest, #342408), anti-CD3 APC (#555335), anti-CD4 FITC (#555346), anti-CD8 FITC (#555634), anti-CD19 FITC, (#555412) anti-CD279 (PD-1) PE (#557946), and anti-CD25 PE-Cy5 (#555433). After 20 min of incubation at room temperature, in the dark, cells were prepared with BD FACS Lysing solution and incubated for 10 min at room temperature in the dark. Cells were then washed twice in PBS for 5 min in 500 RCF. After appropriate staining, samples were analysed using FACSCaliburTM flow cytometer (BD Biosciences, NJ, USA) and CellQuest ProSoftware (Version 6.0). A minimum of 10,000 events were acquired and analysed using CellQuest Software (Version 6.0). 

### 4.3. Compliance with Research Ethics Standards

All patients and parents or legal guardians were informed in detail in oral and written form about the course, aims, and scope of the conducted research. All patients over 16 years and parents or guardians signed an informed consent to participate in the study. The study was carried out in compliance with the Declaration of Helsinki. The study design was approved by the Bioethics Committee at the Medical University of Lublin (KE-0254/236/2020).

### 4.4. Statistical Analyses

Results from measurable parameters are presented as the mean, median, minimum, maximum values and standard deviation. Immeasurable parameters are presented as means of count and percentage. The normal distribution of variables was checked using the Shapiro–Wilk test. Student’s *t*-test and the Mann–Whitney U test were used for intergroup comparisons for normally and non-normally distributed data, respectively. Differences between more than two groups were analysed with the Kruskal–Wallis test, ANOVA, and multiple comparisons of mean ranks (as post hoc analysis) with the Bonferroni correction. The associations between pairs of variables were assessed with Spearman’s rank correlation. Statistical significance was considered at *p* < 0,05. The statistical analysis was carried out using Statistica 13.3 software (StatSoft, Kraków, Poland).

## 5. Conclusions

In conclusion, our study highlights PD-1 overexpression in PIMS-TS/MIS-C pathogenesis. Since the presented findings are a pioneer attempt regarding these correlations, more studies are needed to enhance this knowledge and translate these findings into clinical practice to decrease PIMS-MIS/C morbidity and mortality. We believe that observations of the overexpression of PD-1 with subsequent normalizing percentages of T cells expressing CD279 highlight their role in the pathogenesis of disease as normalization was associated with proper treatment and overall health improvement. The overexpression of the PD-1 receptor during the acute phase might be related to the excessive activation of these cells and emphasize their role in pathogenesis, or, on the other hand, might be the result of an abnormal reaction to previous contact with the virus, constituting one of the possible indicators of lymphocyte exhaustion/anergy, which could potentially be associated with an incorrect response to the viral antigen and then corrected with the proper treatment. Such observations also show that the PD-1/PD-L1 axis is involved during PIMS-TS but not in prolonged responses after diseases. More research is needed to explain this phenomenon.

## Figures and Tables

**Table 1 ijms-25-05968-t001:** Baseline demographic and clinical characteristics of the patients.

Parameters	PIMS-TS (n = 33)	Control (n = 35)	*p* Value
	Median (Range)	Median (Range)	
Age (years)	9.26 (2.0–18.0)	10.0 (2.0–18.0)	0.085
ESR (mm/h)	67.5 (54.0–120.0) *	6.0 (2.0–16.0)	<0.0001
CRP (mg/dL)	8.9 (4.7–22.2) *	0.06 (0.06–0.08)	<0.0001
anti-SARS-CoV-2 IgG (ng/mL)	316.0 (155.0–508.0) *	97.1 (53.2–417.0)	0.09
anti-SARS-CoV-2 IgM (ng/mL)	0.36 (0.26–0.72) *	0.3 (0.16–0.91)	0.21
Treatment		ND	ND
GC	13 (39.4%)		
IVIG	33 (100%)		
ASA	33 (100%)		

* Before treatment (point 0); ESR—erythrocyte sedimentation rate; CRP—C-reactive protein GC—glucocorticosteroids; IVIG—intravenous immunoglobulins, ASA—acetylsalicylic acid, SARS IgG/IgM—antibodies against SARS-CoV-2 IgG/IgM; the comparison of laboratory tests performed during subsequent visits, notably higher activity of inflammatory parameters was observed before taken treatment. Interestingly, the expression of PD-1 protein on CD4+ cells decreased gradually with a minimum percent after 3 months from diagnosis (Table 1).

**Table 2 ijms-25-05968-t002:** Results of laboratory tests during subsequent visits (0—before treatment, I—after 6 weeks, II—after 3 months, III—after 6 months, IV—after 9 months, V—after 12 months from diagnosis).

Parameters	Number of Visits							
	0 (n = 33)	I (n= 33)	II (n = 24)	III (n = 24)	IV (n = 12)	V (n = 8)	Control (n = 35)	
	Median (IQR)							*p* Value
ESR [mm/h]	67.5 (54.0–120.0)	13.5 (8.5–24.5)	14.0 (8.0–17.0)	11.0 (6.5–18.5)	11.5 (7.0–23.5)	15.0 (4.0–20.0)	6.0 (2.0–16.0)	<0.0001
CRP [mg/L]	8.9 (4.7–22.2)	0.06 (0.06–0.15)	0.08 (0.06–0.21)	0.06 (0.06–0.15)	0.09 (0.06–0.2)	0.11 (0.06–0.16)	0.06 (0.06–0.08)	<0.0001
Ferritin [ng/mL]	300.4 (161.9–596.7)	25.3 (15.1–51.2)	26.6 (14.3–39.1)	21.1 (14.6–29.4)	22.0 (12.4–26.4)	23.2 (16.2–33.5)	24.3 (15.2–36.2)	<0.0001
D-dimer [ug/L]	2228 (975–4167)	378.5 (289.5–449.0)	344.5 (240.5–450.0)	292 (206–421)	315.5 (199.5–367.0)	293.0 (213.5–417.5)	354.0 (257–449)	<0.0001
WBC [G/L]	10.7 (7.5–13.6)	6.7 (5.3–8.8)	6.6 (5.1–9.3)	5.8 (5.2–7.8)	5.9 (5.2–7.2)	6.8 (4.8–9.7)	6.0 (4.6–7.2)	<0.0001
Lymphocytes [G/L]	1.7 (0.9–2.7)	2.6 (2.2–3.9)	2.4 (1.8–2.8)	2.2 (1.9–3.2)	2.5 (2.1–2.7)	2.7 (1.8–3.5)	2.2 (1.8–2.5)	0.013
Anti-SARS-CoV-2 IgG	316.0 (155.0–508.0)	155.0 (104.5–411.5)	198.5 (75.0–246.5)	148.0 (62.9–412.0)	141.5 (103.9–262.0)	100.1 (81.2–185.0)	97.1 (53.2–417.0)	0.09
Anti-SARS-CoV-2 IgM	0.36 (0.26–0.72)	0.29 (0.18–0.76)	0.35 (0.19–0.53)	0.21 (0.14–0.40)	0.39 (0.11–0.54)	0.39 (0.3–0.53)	0.3 (0.16–0.91)	0.21
FoxP3+ CD4+ CD25+ [%]	1.0 (0.7–1.3)	1.0 (0.65–1.37)	0.86 (0.7–1.3)	0.87 (0.7–1.2)	0.71 (0.6–1.1)	0.55 (0.4–1.0)	1.16 (0.8–1.6)	0.13
CD4+ PD-1+ T cells [%]	7.4 (4.8–10.4)	4.5 (3.4–6.9)	0.86 (0.7–1.3)	4.3 (3.0–5.7)	3.33 (2.5–6.3)	2.98 (2.3–5.7)	3.95 (2.6–5.7)	0.0065
CD8+ PD-1+ T cells [%]	7.29 (4.4–12.3)	8.1 (4.6–11.2)	3.84 (2.6–6.0)	6.2 (4.4–9.0)	5.14 (2.8–7.1)	4.0 (2.1–7.1)	5.42 (3.4–8.2)	0.057
CD19+ PD-1+ B cells [%]	1.19 (0.7–2.7)	1.6 (1.3–2.7)	1.36 (0.9–2.1)	1.3 (0.8–2.5)	1.39 (0.8–2.7)	1.0 (0.5–1.4)	1.54 (0.9–1.0)	0.27

Expression of the PD-1 receptor on lymphocytes CD8+, CD4+ and CD19+. ESR = erythrocyte sedimentation rate, CRP = C-reactive protein, WBC = white blood cells, FoxP3+ CD4+ CD25+ T regulatory cells, CD4+ T helper cells, CD8+ T cytotoxic cells, and CD19+ B cells.

## Data Availability

Data are contained within the article.

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
