# Peer review of "Programmed Cell Death Protein-1 Regulation in Response to SARS-CoV-2 in Paediatric Multisystem Inflammatory Syndrome Temporally Associated with SARS-CoV-2: A Prospective Cohort Study"

_ijms, 2024, doi:10.3390/ijms25115968_

Round 1

Reviewer 1 Report

Comments and Suggestions for Authors

The authors described the PD1 findings in T cells of patients with COVID infection diagnosed to have PIMS, and concluded that PD1 play a role in the pathogenesis of PIMS in cases with COVID infection.

These are my comments:

Which comparison is the p value in table 2 represents? Is it between control and at Stage 0 (or the mean of all stages)?

There are only 8 cases at one year follow up (stage 5). Is it because others have already recovered and these 8 cases continue to be unwell? Is so, these cases are more severe and therefore one would expect the findings to be worse. It might be interesting to look at just these 8 cases on the other stages. How it is deferred from those who recovered earlier.

Table 1, the title is demographic and clinical characteristics. However, only age and biochemical parameters were described. Suggest to add other information such as gender, any comorbid etc.

It will be meaningful to know which 13 cases were treated with steroid. Was the 8 cases follow up at one year treated with steroid?

In table 1, should also include which organs/ tissues were involved by PIMS?

Was lung involved in all the cases? If so, what are their chest x-ray findings?

Please provide the reference to the diagnostic criteria of PIMS.

In table 2, what is FoxP3+ CD4+ CD25 T cells? What is the different between this cell with CD4+ T cells?

CD4+ PD1+ T cells was the same between controls and stage 1 to 5. The only difference was between Stage 0 and control. I think this is important observation. Post Covid period did not have a raised PD1+ CD4 cells. Does this still mean PD1 play a role in PIMS? Please explain this.

What is the percentage of PD1 positive T cells?

The results were poorly described. Many of the data in table 2 was not mentioned in the text.

In the methodology, flow cytometry, antibodies used need to have more information. For example, what is anti-14 PE? Where are these antibodies obtained?

The number of cases (33) and controls (35) were not enough to provide a good statistical analysis. Please provide a sample size calculation to support that it is sufficient for the statistical analysis. Moreover, the number markedly reduced on the follow up cases.

The conclusion is not entirely supported. The number of cases is low and the correlation is not strong. I agree more studies are needed.

Comments on the Quality of English Language

None

Author Response

Dear Reviewer,

 Thank you for your valuable comments to our manuscript. Please find the detailed point-by-point answers to Reviewers suggestions and questions.

 The authors described the PD1 findings in T cells of patients with COVID infection diagnosed to have PIMS, and concluded that PD1 play a role in the pathogenesis of PIMS in cases with COVID infection.

These are my comments:

Which comparison is the p value in table 2 represents? Is it between control and at Stage 0 (or the mean of all stages)?

Re: The analysis in Table 2 was calculated with the use of Kruskal Wallis test ANOVA to compare tested parameters examined during all tested visits.

There are only 8 cases at one year follow up (stage 5). Is it because others have already recovered and these 8 cases continue to be unwell? Is so, these cases are more severe and therefore one would expect the findings to be worse. It might be interesting to look at just these 8 cases on the other stages. How it is deferred from those who recovered earlier.

RE: Subsequent visits with control of laboratory tests included patients who came for a visit on the date specified in the plan. Therefore, the last follow-up in this study included only 8 patients. The patients were in good condition, no disease symptoms were observed in the medical history and physical examination.

In general, further follow-up was recommended for all children after PIMS-TS.

Table 1, the title is demographic and clinical characteristics. However, only age and biochemical parameters were described. Suggest to add other information such as gender, any comorbid etc.

It will be meaningful to know which 13 cases were treated with steroid. Was the 8 cases follow up at one year treated with steroid?

RE: Patients were treated in accordance with Polish recommendations [11]

First-line treatment was intravenous immunoglobulin (IVIG) infusion at a dose of 2.0 g/kg.

Indications for the use of glucocorticoids included:

  1. a) severe or deteriorating general condition of the child,
  2. b) characteristics of the shock,
  3. c) lack of improvement, in particular persistent fever, 24-36 hours after the end of the IVIG infusion,

No child experienced coronary artery aneurysms or anaphylactic reactions after receiving IVIG.

Treatment with glucocorticoids was used for 3 to 6 weeks.

Tab. 1: 1st examination: study group: 19 were male control group: 16 male

In table 1, should also include which organs/ tissues were involved by PIMS?

Was lung involved in all the cases? If so, what are their chest x-ray findings?

RE: The patients had systemic symptoms of PIMS-TS, including: the gastrointestinal tract, the cardiovascular system, the nervous system, the respiratory system, the mucous-cutaneous system, the kidneys, and signs of coagulopathy. Due to the diversity of manifestations, we did not analyze the clinical symptoms in this study.

Please provide the reference to the diagnostic criteria of PIMS.

Patients were diagnosed  in accordance with current criteria  [10,11].

In table 2, what is FoxP3+ CD4+ CD25 T cells? What is the different between this cell with CD4+ T cells?

CD4+ PD1+ T cells was the same between controls and stage 1 to 5. The only difference was between Stage 0 and control. I think this is important observation. Post Covid period did not have a raised PD1+ CD4 cells. Does this still mean PD1 play a role in PIMS? Please explain this.

What is the percentage of PD1 positive T cells?

RE: CD3+CD4+CD25+FoxP3+ cells are T regulatory (Treg) cells, as stated in the Table, while CD3+CD4+ cells are T helper (Th) cells. Treg cells act to suppress the immune response, whereas Th cells activate other cells. Please clarify your questions so we can address your concerns properly. We are unsure if you want us to discuss the differences between Tregs and Th cells in the context of PIMS-TS or in general health. However, we have added appropriate notes below the table to clearly identify the selected lymphocyte populations.

We believe that the observation of overexpression of PD-1, followed by subsequent normalization of T cells expressing CD279, highlights their role in the pathogenesis of the disease, as normalization was associated with proper treatment and overall health improvement. Overexpression of the PD-1 receptor during the acute phase might be related to the excessive activation of these cells, emphasizing their role in pathogenesis. Alternatively, it might result from an abnormal reaction to previous contact with the virus, constituting a possible indicator of lymphocyte exhaustion or anergy, which could potentially be associated with an incorrect response to the viral antigen and then corrected with proper treatment. More research is needed to explain this phenomenon. These observations also show that the PD-1/PD-L1 axis is involved during PIMS-TS but not in the prolonged response after the disease. We have added this information to the text.

We did not assess the expression of PD-1 on all CD3 positive cells but rather on their main subpopulations. This is a more efficient way to understand the pathomechanism because each subpopulation plays a different role in the immune response.

The results were poorly described. Many of the data in table 2 was not mentioned in the text.

In the methodology, flow cytometry, antibodies used need to have more information. For example, what is anti-14 PE? Where are these antibodies obtained?

RE: We decided to refrain from discussing each of the obtained results in detail because observations regarding ESR, CRP, or leukocytosis in patients are not the subject of this work and are well-established in the context of PIMS-TS. In our opinion, a description of each element of the table would introduce unnecessary complexity, and the table itself serves as a clear and interpretable graphical presentation of the results. We have added the appropriate catalog numbers and information indicating that the antibodies are fluorochrome-conjugated. Moreover, this paper presents PD-1 expression on selected cell populations. Perhaps the Reviewer was confused by the mention of CD45 and CD14 antibodies in the “Materials and Methods” section, which is why we included a gating strategy. The gating strategy involves first gating mononuclear cells (CD45+/CD14+), then lymphocytes expressing CD3 (T cells) or CD19 (B cells), and later CD4+ or CD8+ among CD3+ T cells. Subsequently, we gate CD4+PD1+ among CD3+CD4+ cells and CD8+PD1+ among CD3+CD8+ cells. To clarify, we did not include all the data in the table; therefore, we have added additional information about the gating method to clearly explain the reason for using the selected antibodies.

The number of cases (33) and controls (35) were not enough to provide a good statistical analysis. Please provide a sample size calculation to support that it is sufficient for the statistical analysis.

RE: It can be concluded that the prevalence of PIMS-TS is estimated to be at 1 case per 1000 children infected by SARS-CoV-2. After calculation, the minimum number of necessary samples to meet the desired statistical constraints is 3. We calculated the margin of error for group of 33 patients with PIMS diagnosis in relation to app. 1,120,000 children diagnosed with Covid-19 in Poland in that year which came to 3.64%.

Moreover, the number markedly reduced on the follow up cases.

We have tried to explain above.

The conclusion is not entirely supported. The number of cases is low and the correlation is not strong. I agree more studies are needed.

RE: Thank you, we also believe that further studies are needed.

We do hope that after corrections the paper can meet your requirements.

Best regards,

Reviewer 2 Report

Comments and Suggestions for Authors

Overall a good study to highlight the importance of the expression of PD-1 gene in the multi systemic inflammatory syndrome associated with COVID-19 infection. 

Any comments on the pathogenesis of MIS in pediatric population who already have immune conditions?

I understand this study was done during the first wave of the COVID infection when there was no widespread availability of the COVID vaccine among the pediatric population was not there but now was there any data identified which points to the increase /decrease of MIS among the vaccinated pediatric population.

Author Response

Dear Reviewer,

 Thank you for your valuable comments to our manuscript. Please find the detailed point-by-point answers to Reviewers suggestions and questions.

Overall a good study to highlight the importance of the expression of PD-1 gene in the multi systemic inflammatory syndrome associated with COVID-19 infection. 

Any comments on the pathogenesis of MIS in pediatric population who already have immune conditions?

According to current knowledge, the symptoms of PIMS-TS occur as a result of both humoral and cellular disturbed immune responses. Cytokine profiling in PIMS-TS patients indicates myeloid cell chemotaxis and inflammation. An assessment of cytokines showed an increased release of the pro-inflammatory cytokines Il-1, IL-6, IL-18, IL-17A,  TNF-α and the chemokines, which regulate the recruitment and modulation of natural killer (NK) cells, neutrophils, monocytes and T lymphocytes.  Pro-inflammatory cytokines play a key role in the pathogenesis of disease.

Nevertheless, the pathogenesis of PIMS-TS requires answering many questions

 RE: A detailed individual and family medical interview was conducted. No previous diseases associated with immune response disorders were detected.

I understand this study was done during the first wave of the COVID infection when there was no widespread availability of the COVID vaccine among the pediatric population was not there but now was there any data identified which points to the increase /decrease of MIS among the vaccinated pediatric population.

RE: The study was done during the first wave of the COVID infection when there was no widespread availability of the COVID vaccine among the pediatric population.

We hope that after the corrections the paper will meet the requirements.

Best regards,

Round 2

Reviewer 1 Report

Comments and Suggestions for Authors

The authors have answered the concerns in the revised manuscript. 

Comments on the Quality of English Language

No further comment